# A Pulsed Electric Field Accelerates the Mass Transfer during the Convective Drying of Carrots: Drying and Rehydration Kinetics, Texture, and Carotenoid Content

**DOI:** 10.3390/foods12030589

**Published:** 2023-01-30

**Authors:** Si-Yeon Kim, Byung-Min Lee, Seok-Young Hong, Hyun-Ho Yeo, Se-Ho Jeong, Dong-Un Lee

**Affiliations:** Department of Food Science and Technology, Chung-Ang University, Anseong 17546, Republic of Korea

**Keywords:** carotenoid, drying, rehydration, kinetics, pulsed electric field (PEF)

## Abstract

The pulsed electric field (PEF) is a non-thermal food processing technology that induces electroporation of the cell membrane thus improving mass transfer through the cell membrane. In this study, the drying and rehydration kinetics, microstructure, and carotenoid content of carrot (*Daucus carota*) pretreated by PEF during convective drying at 50 °C were investigated. The PEF treatment was conducted with different field strengths (1.0–2.5 kV/cm) using a fixed pulse width of 20 µs and at a pulse frequency of 50 Hz. The PEF 2.5 kV/cm showed the shortest drying time, taking 180 min, whereas the control required 330 min for the same moisture ratio, indicating a 45% reduction in drying time. The rehydration ability also increased as the strengths of PEF increased. PEF 2.5 kV/cm resulted in 27.58% increase in moisture content compared to the control after rehydration (1 h). Three mathematical models were applied to the drying and rehydration data; the Page and Peleg models were selected as the most appropriate models to describe the drying and rehydration kinetics, respectively. The cutting force of the sample was decreased as the strength of PEF increased, and a more homogeneous cellular structure was observed in the PEF pretreatment group. The reduction in drying time by PEF was beneficial to the carotenoid content, and PEF 2.5 kV/cm showed the highest preservation content of carotenoid. Overall, these results suggested that the pretreatment of PEF and the drying and rehydration rate influence the quality of products, functional components, and cellular structure.

## 1. Introduction

The carrot (*Daucus carota*) is known as the most commercial root vegetable because of its multiple types of consumption (dried, rehydration) and is mainly cultivated in Southwestern Asia, Europe, and Africa [1]. The statistics of the Food and Agriculture Organization of the United Nations (FAO) reported that the production quantity of carrots and turnips was 59 million tons in 2020 worldwide [2]. Carrots contain 86–89% water, 7% carbohydrates, 3% fiber, including 50–92% cellulose and hemicellulose, approximately 4% lignin, minor proteins, fat, minerals, and antioxidant compounds, especially carotenoids [3]. However, rapid contamination by microorganisms occurs because of the high moisture content, nutritional component, pH (6.0–6.5), and the high initial total viable microorganism (10^5^–10^6^ CFU/mL) content [4]. Therefore, to increase the shelf life and the preservation of unprocessed carrots, thermal processing is required in the food industry [5].

Convective drying is the most economical thermal technology which is uses passing air and a regulated relatively high temperature [6]. However, during air-drying, moisture in the outer layer of food is initially removed before internal moisture is removed which causes undesirable physical effects such as case hardening, crust formation, and irreversible structural changes [7]. Furthermore, phytochemicals including carotenoids, lycopene, lutein, and vitamins in carrots are reduced due to the isomerization and degradation by heat treatment during hot-air drying [8]. Especially, carrots have a relatively complex anisotropic cellular structure, and heterogeneous structural regions (cortex, core) with vascular tissue (xylem, phloem) which inhibit the mass transfer through the cell membrane during drying and rehydration; thus, the drying process requires a long time compared to other vegetables [1]. Therefore, to minimize the negative effect during convective air drying, a reduction in drying time is necessary to preserve the final product’s structural, nutritional, and rehydration quality.

There have been numerous studies accelerating the rate of mass transfer during convective drying, and different methods were developed, applied, and evaluated, such as radio frequency-assisted hot air drying [9], infrared blanching [10], ultrasound, and pulsed electric field [11], and pulsed electric field [12]. Among the emerging technologies, pulsed electric field (PEF), which induces reversible and irreversible electroporation on the cell membrane, could increase the mass transfer during the drying process effectively with limited deterioration of the cellular structure which could decrease the rehydration rate [13]. Pulsed electric field (PEF) has been proposed for thermally sensitive food materials due to the damage to the anatomical integrity of vegetable cells which can facilitate moisture diffusivity during food processing [14]. The application of PEF on carrots improves drying and rehydration quality, whereas few other methods address the relationship between water leaks and sorption properties, texture, cellular structure, and carotenoid contents. This study is devoted to investigating the moisture transfer rate during drying and carotenoid contents by modelling the water leaks and sorption. Moreover, there are no standard pilot-scale PEF application methods for the convective drying of carrots, thus suggesting the standard condition of PEF with various strengths, the texture properties, cellular structure, and carotenoid contents should be further investigated.

Consequently, the aim of this study was to investigate the effects of PEF on the convective drying of carrots. Evaluation of the drying and rehydration qualities, drying and rehydration kinetics, texture properties microstructure, and carotenoid content preservation were also observed.

## 2. Materials and Methods

### 2.1. Sample Preparation and PEF Treatment

Carrots (*Daucus carota* L.) were obtained from a local market (Anseong, Republic of Korea) then peeled and stored in a refrigerator (4 °C) for 12 h. After that, carrots were cut into 10 mm × 10 mm × 10 mm and the samples were treated with field strength values of 1.0, 1.5, 2.0, and 2.5 kV/cm by setting out voltage (%) values to 25, 40, 55, and 70%. A total of 60 g of carrots was immersed in an 80 mm chamber and then placed between the two electrodes parallel to the electric current flow. The electric switch of IGBT (Insulated Gate Biplar Transistor) and the condition of the electrical connection were the following: power supply voltage, 400 V; rated current, 25 A; mains frequency, 50 Hz; and type of connection, 3-phases (3P + PE). The pulse number, the pulse width, and the frequency (Hz) were 500, 20 μs, and 50 Hz, respectively, with a 5-kW pulse generator used (HVP-5, DIL, Quakenbruck, Germany).

### 2.2. Electrical Conductivity and Cell Disintegration Index (Z-Index)

Biological electrical conductivity *σ* (S/m) was measured by using an LCR meter (LCR-8000G, Gwinstek, Tucheng, Taiwan), in accordance with Angersbach et al. [15]. To measure the electric conductivity of the sample, it was cut into 10 mm × 10 mm × 10 mm and measured with a range from 1 kHz to 2 MHz, before being calculated by using Equation (1).
(1)$$\sigma \left(\omega^{s}\right)=\frac{1}{A\left|Z{\left(j\omega \right)}^{s}\right|}$$
where 1, *A*, and $Z{\left(j\omega \right)}^{s}$ are the length of sample, area horizontal to the electrical field, and system impedance, respectively. To evaluate the Z-index, the untreated sample was freeze for 24 h and thawed at 25 °C for 1 h; we measured the electric conductivity using an LCR meter. All samples were measured and calculated as described in Lebovka et al. [16].

### 2.3. Drying Kinetics Modelling

The size of the sample was prepared with 10 mm × 10 mm × 10 mm and the initial weight was 3.5 g which was then dried at 50 °C using a convective dryer (SFC-203, Shinsaeng, Paju, Republic of Korea). The total time of drying was 6.5 h, and the weight was measured every 30 min. Each weight of the sample was transformed into moisture ratios (MR) and then fitted to the three mathematical models (Newton, Page, Henderson and Pabis), as described in Biswas et al. [17] with slight modifications. To evaluate the goodness of fit, correlation coefficient ($R^{2}$), root mean square error (RMSE), and the sum of squares for residuals (SSE) were calculated following Equations (2)–(4)
(2)$$R^{2}=1-\frac{\sum_{i\;=1}^{N}{\left(MR_{pre,i}-MR_{exp,i}\right)}^{2}}{\sum_{i\;=1}^{N}{\left(MR_{exp,i}-{\bar{MR}}_{exp}\right)}^{2}}$$
(3)$$\mathrm{RMSE}=\sqrt{\frac{1}{N}\sum_{i=1}^{N}{\left(MR_{pre,i}-MR_{exp,i}\right)}^{2}}$$
(4)$$\mathrm{SSE}=\sum_{i=1}^{N}{\left(MR_{exp,i}-MR_{pre,i}\right)}^{2}$$
where $MR_{pre}$ is the predicted moisture ratio, $MR_{exp}$ is the experimental moisture ratio, ${\bar{MR}}_{exp}$ is the standard of experimental moisture ratio and N is the number of observations. The higher value of $R^{2}$ and the lower values of RMSE and SSE indicate the goodness of fit.

### 2.4. Rehydration Kinetics Modelling

To determine the rehydration rate of carrots, the samples were rehydrated using a water bath at 90 °C for 2 h. The sample was weighed every 10 min and the kinetics used in the rehydration process were fitted to the three empirical models (Peleg, Exponential, First-order) according to Lopez-Quiroga et al., (2019). The equation used in the Peleg model is as described by Peleg [18] and García-Pascual et al. [19] and the correlation coefficient ($R^{2}$), root mean square error (RMSE), and mean relative error (MRE) were calculated to determine the goodness of fit. The mean relative error (MRE) was calculated by Equation (5)
(5)$$\mathrm{MRE}=\frac{100}{N}\sum_{i=1}^{N}\left|\frac{W_{exp,i}-W_{pre,i}}{W_{exp,i}}\right|$$
where *W_exp_* is the moisture content of the sample at time t, W_pre_ is the predicted moisture content, and N is the number of experiment data. The higher value of *R*^2^ and the lower values of RMSE and MRE indicate the goodness of fit.

### 2.5. Texture

To assess the cutting force, the size of the sample was prepared with 10 mm × 10 mm × 10 mm, and then measured using a Texture Analyzer (TAHDi/500, Godalming, UK). We used a Warner–Bratzler reversible blade and a 50-kg power cell was used for the measurements based on the maximum shear force (N). Pre-test speed was set at 10 mm/s, test speed 4.00 mm/s, post-test speed 10 mm/s, and distance at 30 mm.

### 2.6. Microstructure

To confirm the microstructure of the sample, a scanning electronic microscope (S-3400N, Hitachi High Technologies Co., Tokyo, Japan) was used. Before measurement, the sample was dried in a desiccator for at least 12 h and coated with platinum (Pt-Pb). The SEM picture was observed at ×500 magnification at 10 kV.

### 2.7. Total Carotenoid Content

To measure the total carotenoid content of carrots, the method described by Sadler et al. [20] was used. A total of 0.5 g of dehydrated and rehydrated carrot sample was mixed with 0.5 mL ${\mathrm{Cacl}}_{2}\cdot 2H_{2}O$ + 25 mL extraction solution (Hexane:Acetone:Ethanol = 50:25:25) + BHT 0.01% and left at 4 °C for 20 min. After that, we added 7.5 mL of reagent-grade water and left it at 4 °C for 10 min. Then, the absorbance of the supernatant was measured at 450 nm using a UV spectrophotometer. We substituted the measured absorbance value into the Equation (6) below to obtain total carotenoid (μg/g dry matter).
(6)$$\left[cc\left(\frac{\mu g}{g}\right)\right]=\frac{A_{450}\times 536.85\times V}{m\times 137.4}$$

$A_{450}$ represents the absorbance value at 450 nm of the sample, 536.85 represents the carotenoid molecular weight, *V* represents the volume, *m* represents the sample weight, and 137.4 represents the carotenoid extinction coefficient. Each sample was measured in triplicate.

### 2.8. Statistical Analysis

All experiments were performed in triplicate and all the values were represented as mean ± standard deviation. The statistical analysis was performed using SPSS.26 (IBM Corp., Armonk, NY, USA) according to Duncan’s multiple range test at *p* = 0.05. Graphs were made using GraphPad Prism (v.8.0.1, GraphPad Software Co, Ltd., Boston, MA, USA).

## 3. Results and Discussion

### 3.1. Electric Conductivity and Z-Index

The results of electric conductivity were increased as the strengths of PEF increased; thus, 2.5 kV/cm was the highest among samples (Figure 1). The results of the PEF treatment of peppers [21] were consistent with this study in that as the higher strengths of PEF were applied, the deterioration of the cell membrane increased; thus, the value of electric conductivity increased. Teissié et al. [22] reported that an increase in cell permeability which showed higher electric conductivity was led by the external electric field on the cell membrane. This polarization effect which associated with the diffusion, mass transfer, and heat transfer of the food products controlled by pulse number, duration of the pulse, the strength of the field, cell type, and cell structure [23]. As the strengths of the field increased, the membrane rupture moment was reduced and the pore-widening rate and pore size on the cell membrane increased, thereby increasing the membrane permeabilization [24]. To induce reversible or irreversible electroporation on the cell membrane, certain threshold strengths of the field, especially more than 0.3 kV/cm, should be exceeded for plant tissue [25]. The values of the Z-index which indicate the level of electroporation were 0.26, 0.57, 0.73, and 0.93, respectively, for PEF1.5, PEF2.0, and PEF2.5, which is about 3.5 times higher than that of 1.0 kV/cm. A similar trend was also reported by Amiali et al. [26], where an increase in electric field strength on arils showed a higher value of the Z-index in 1.5 kV/cm (0.8) than in 0.5 kV/cm (0.6) with the same pulse number (150). A previous study by Jeong et al. [27] showed that the highest value of the Z-index was observed in the highest field strength (2.0 kV/cm). Regarding the results of electric conductivity and Z-index, the effects of PEF on the increased cell membrane permeability by the strengths of PEF which can strongly contribute to the rate of mass transfer during food processing may be explained.

### 3.2. Drying Kinetics Discrimination

The drying time was decreased in PEF-treated group and as the strengths of PEF increased, the drying process was accelerated (Figure 2). The moisture content had a time-dependent decrease until it reached below 10%. The drying time of the sample ranged from 180 min to 330 min. Equilibrium moisture content was reached after 5.5 h (control), 4 h (PEF1.0), 3.5 h (PEF1.5), and 3 h (PEF2.0 and PEF 2.5), respectively, and approximately more than 45% drying time was reduced compared to the control. At the same time, after 4 h, the moisture content of PEF 2.5 kV/cm reached 10%; however, the moisture content of the control was 51%. Furthermore, due to the improvement in mass transfer of PEF-treated group, moisture of the interior cell was primarily removed at the beginning of drying; therefore, the case hardening phenomenon and excessive shrinkage were inhibited. A similar tendency was found by Liu et al. [28] in the drying of potatoes, where through the use of PEF, drying time was strongly decreased. Moisture can leak out from the tissue by transmembrane transport which passes through the tonoplast and plasmalemma and symplastic transport which pass the cytoplasm and the apoplastic cell wall [29]. PEF induces electroporation of the cell membrane and the vacuolar membrane this weakens the barrier between the vacuole and cytoplasm improving water diffusion [30]. Weaver and Chizmadzhev. [31] reported that, as PEF increased, more water is lost (even bound water in the cell wall) due to the permeability of the cell membrane. Furthermore, the permeability of the cytoplasm membrane, which plays a major role as a regulator of mass transfer, increased, thus enhancing the diffusion and migration of small molecules [32].

The results of mathematical fitting and the values of parameters using the three models were presented in Table 1. Regarding model constant (*k*, a, n) and parameter for evaluating the goodness of fit ($R^{2}$, RMSE), the Page model revealed the best fit for drying carrots. The $R^{2}$ for the Page model was between 0.979 and 0.999. The arrangement of RMSE and SSE were 0.005 to 0.030 and 0.001 to 0.007, respectively. The value of *k* was significantly increased to 0.602, 0.738, 0.920, 1.185, and 1.349 with strengths of PEF (*p* < 0.05). The value of *k* is related to the drying rate and the value of n is associated with water diffusion during the drying process. A value of n more than one means sub-diffusion process and less than one means super-diffusion. The value of n was observed as being more than one in all samples. Regarding the results of Figure 2 and Table 1, it can be concluded that pretreatment of the PEF group could enhance the moisture diffusivity during the drying process, thus removing water from the interior of the cell faster than the untreated sample.

### 3.3. Rehydration Kinetics Discrimination

The rehydration kinetics curve of the sample is illustrated in Figure 2. Rehydration capacity was gradually increased with an increase in the strength of PEF. From the results, it was confirmed that the rehydration ability increased as the field strength of PEF increased, compared to the control. Among them, the time with the highest difference in weight was 60 min after rehydration and was 27.58% higher in PEF 2.5 kV/cm compared to the control. The moisture content of the control after rehydration was 27.58% lower than PEF 2.5 kV/cm. Such results were also reported by Lammerskittenet al. [33]. They showed that treatment with PEF improved the rehydration capacity of apples and that the water content of rehydrated apples was almost recovered compared to fresh apple tissue after 1 h, whereas the untreated sample achieved lower than 30% of this value. Among the three mathematical models, Peleg was most suitable for describing the experimental data in this research (Table 2). The values of *k*_1_ were control (0.0815), PEF 1.0 kV/cm (0.0800), PEF 1.5 kV/cm (0.0757), PEF 2.0 kV/cm (0.1204), PEF 2.5 kV/cm (0.1198), respectively. The *k*_2_ of the control (0.1311) was higher than the PEF-treated group (0.0800–0.1077). Decreases in the value of *k*_1_ and *k*_2_ indicate an increase in initial water absorption rate and a higher equilibrium moisture content, respectively. Such behavior is consistent with a previous study [26], where the value of *k*_1_ and *k*_2_ decreased with the strengths of PEF and thermal treatment on carrots. Rehydration ability is used to evaluate the deterioration of the cellular structure of dried products [33]. The capacity of rehydration promoted by an intact cellular structure, especially with a larger internal space and widened tissue, enhances water absorption during the rehydration process [26]. Liu et al. [28] found that the application of PEF enhanced the rehydration ability, thereby promoting the rate of cellular structure restoration which showed a polyhedral shape. The current study found that pretreatment with PEF improves the movement of water through the cell membrane, which helps to preserve the integrity of the cellular structure compared to untreated samples that experience severe shrinkage due to rapid water loss.

### 3.4. Cutting Force

The textural property was determined by the value of cutting force and the results are shown in Figure 3. Pretreatment of PEF reduced the cutting force of the sample, thus the lowest value of cutting force was obtained in PEF 2.5 kV/cm. The cutting force of rehydrated sample was significantly decreased as the strengths of PEF increased (*p* < 0.05). The value of the control, PEF 1.0 kV/cm, PEF 1.5 kV/cm, PEF 2.0 kV/cm, PEF 2.5 kV/cm were 53.82 ± 7.87 N, 47.78 ± 7.13 N, 42.44 ± 5.61 N, 41.79 ± 8.22 N, 37.24 ± 5.78 N, respectively. The cutting force of a carrot depends on the cortex region and endodermis region which determine the required power of cutting in this study; the cortex region was used to measure the cutting [34]. Disruption of cell membrane induce the loss of turgor pressure which reduces the hardness of the vegetable, thus in the current study, the decrease in cutting force of carrot influenced by the rupture of the cell membrane. This is consistent with Liu et al. [35], where treatment with higher PEF field strengths led to lower firmness and approximately a 30% reduction in firmness was observed. Leong et al. [34] reported similar trends, where the cutting force was decreased by pulse frequency (20–80 Hz), pulse number (500–3000 pulses), and specific energy input (3–113 kJ/kg); thus, improvement in softening of carrots was observed. The “softening effect” when drying potatoes and apples pretreated by PEF is also explored in [33,36]. In this study, the results may be explained by the “softening effect”, which is induced by the cell disintegration by PEF.

### 3.5. Microstructure

As shown in Figure 4, a widened cell wall structure and an increase in pore size and number were observed in PEF pretreated group. However, the control showed significant shrinkage which is induced by collapse of cellular structure [37]. Excessive shrinkage may have a negative influence on the appearance, structure, texture, and flavor [37,38]. Shrinkage is an important parameter of dried products induced by the loss of water, as food polymers cannot maintain cellular structure [39]. An increased deterioration of cell wall structure forms numerous small cavities and causes nonuniform shrinkage that may lead to a decrease in molecular diffusivity during the drying and rehydration process [40]. Plant tissue can maintain the cell wall integrity which is comprised of lignin, cellulose, pectin, and other compounds, especially pectin which is highly correlated with the texture properties during rehydration; thus, an intact cellular structure could accelerate the rehydration rate and give a softer texture [37]. Liu et al. [28] reported that the microstructure of rehydrated potatoes treated by PEF showed a more homogeneous cellular structure; however, in all groups, polyhedral shape was lost by drying. Telfser et al. [41] reported that pretreatment of PEF decreases the deformation of cellular structure when applied during air drying; however, there is no significant difference when applied during vacuum and freeze-drying of basil leaves. Application of PEF accelerate the moisture diffusivity during drying process which makes more porous and homogenous cellular structure, and these properties contribute to the rehydration capacity.

### 3.6. Total Carotenoid Content

The results of PEF on total carotenoid content are illustrated in Figure 5. The total carotenoid contents significantly increased in dried and rehydrated carrots (*p* < 0.05). The values of the control, PEF 1.0 kV/cm, PEF 1.5 kV/cm, PEF 2.0 kV/cm, and PEF 2.5 kV/cm were 26.15 ± 5.12 μg/g, 37.82 ± 1.87 μg/g, 44.44 ± 1.90 μg/g, 50.03 ± 1.81 μg/g and 61.12 ± 5.84 μg/g, respectively, in dried carrot. Similarly to the results from the dried carrot, the values of the control, PEF 1.0 kV/cm, PEF 1.5 kV/cm, PEF 2.0 kV/cm, PEF 2.5 kV/cm were 28.63 ± 4.50 μg/g, 56.05 ± 4.26 μg/g, 66.79 ± 4.71 μg/g, 73.50 ± 1.95 μg/g, and 79.86 ± 2.71μg/g, respectively, in rehydrated carrot. The difference in carotenoid content of rehydrated carrots between the control and 2.5 kV/cm was more than 2.5 times. Carotenoid is in the chromoplast which is inside the cell membrane; thus, it is hard to extract because of multiple barriers such as the cell membrane and cell wall [42]. López-Gámez et al. [43] reported that due to the formation of pores on the cell membrane and the change in cellular structure, which was caused by PEF, carotenoid extraction content increased. Wiktor et al. [44] showed that the application of 3–5 kV/cm pretreatment of PEF increases the total carotenoid contents due to the disruption of cellular structure and ROS (Reactive oxygen species) generation by the stress-response mechanism of the plant. Furthermore, after treatment of PEF (five pulses, 3.5 kV/cm, 0/61 kJ/kg), carotenoid content increased to more than 80% compared to untreated carrots [43]. Parniakov et al. [45] reported that a PEF-treated group has higher carotenoid content than the untreated because of reduced time for exposure to hot air and oxygen at higher temperatures, thereby decreasing the rate of oxidation. Thus, the preserved carotenoids increased by pretreatment of PEF due to the reduction of drying time during the drying process.

## 4. Conclusions

In this study, the effects of PEF with different strengths on drying and rehydration kinetics, texture properties, microstructure, and carotenoid content during convective drying were investigated. Better moisture transfer of the PEF-pretreated group was proved by assessing the drying and rehydration kinetics. The drying and rehydration kinetics curve proved that a PEF pretreatment enhances moisture diffusivity, which results in a reduction in time of drying and rehydration. Various mathematic models were applied to describe the drying and rehydration behavior of carrots. The best-fitting models for drying behavior and rehydration behavior were Page and Peleg, respectively, with higher values of *R*^2^ and lower values of SSE, and MRE. Due to the effective moisture diffusivity, shrinkage was inhibited in PEF pretreated group; thus, more pore numbers and bigger pore sizes were observed. These properties influenced the cutting force of the carrot so that in the PEF-treated group gradually decreased. Furthermore, the effect of pretreatment of PEF on carotenoid content was found to be significant due to the increase in cell permeability and ROS stress response induced by PEF. These results indicate that PEF treatment can serve as a pretreatment method to accelerate the moisture transfer during drying and rehydration and to modify the physicochemical properties of dried carrot, which include tissue cutting force, cellular structure, and carotenoids contents, by inducing electroporation on cell membrane.

Regarding the overall results of this study, PEF could be a promising technology for drying carrots with no excessive deterioration of cellular structure, thus producing high-quality dried and rehydrated carrots.

## Figures and Tables

**Figure 1 foods-12-00589-f001:**
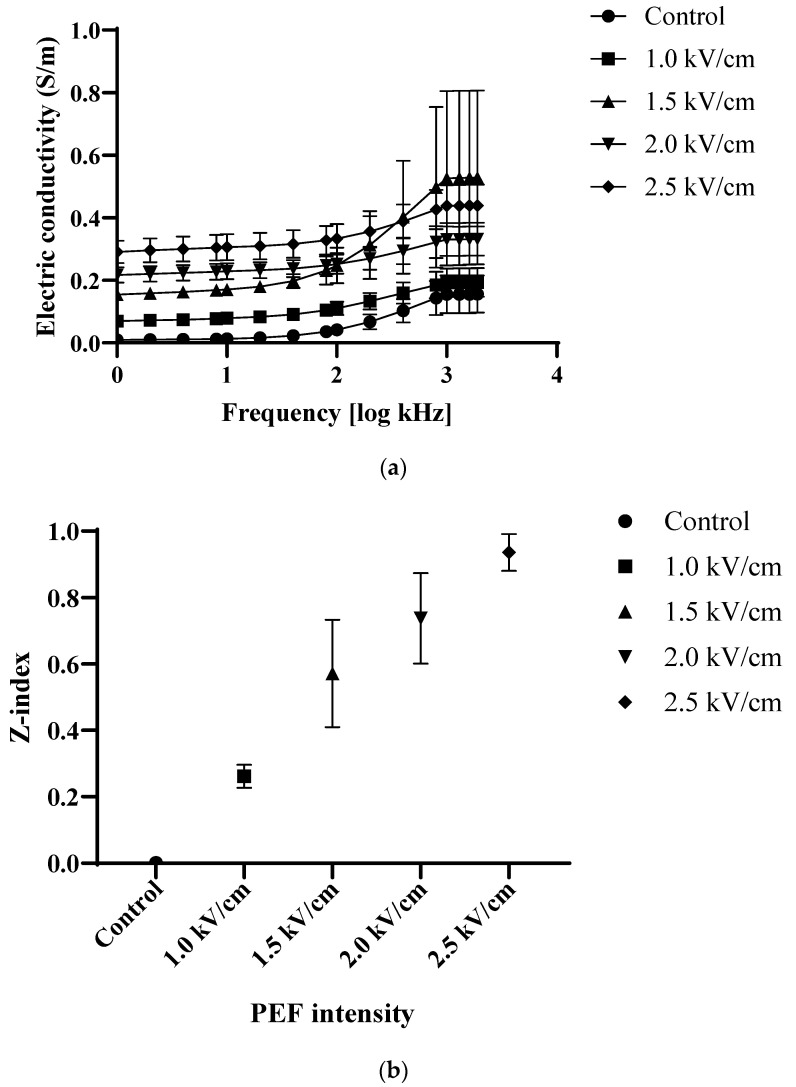
Electric conductivity (**a**) and Z-index (**b**) of carrot with different strengths of PEF pretreatment. Control: untreated; 1.0 kV/cm: carrot treated with PEF 1.0 kV/cm, 1.5 kV/cm: carrot treated with PEF 1.5 kV/cm; 2.0 kV/cm: carrot treated with PEF 2.0 kV/cm; 2.5 kV/cm: carrot with PEF 2.5 kV/cm (n = 3).

**Figure 2 foods-12-00589-f002:**
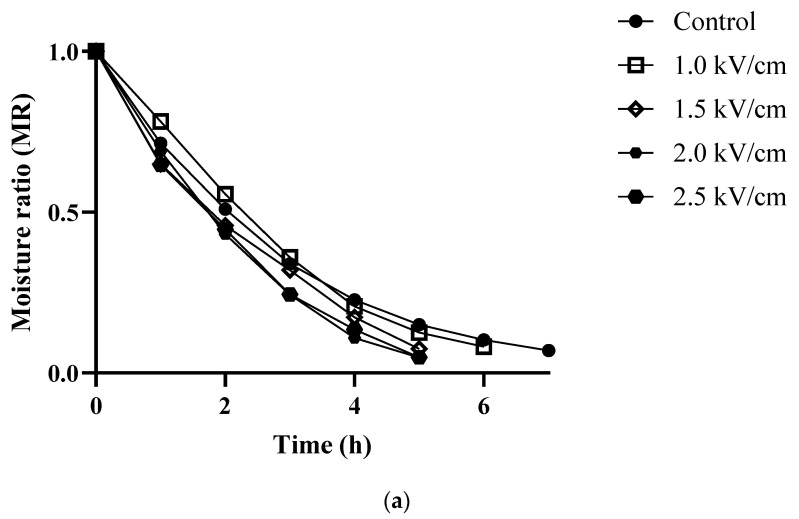
Moisture ratio (MR) of carrot (**a**): drying, (**b**) rehydration with different strengths of PEF pretreatment. Control: untreated; 1.0 kV/cm: carrot treated with PEF 1.0 kV/cm, 1.5 kV/cm: carrot treated with PEF 1.5 kV/cm; 2.0 kV/cm: carrot treated with PEF 2.0 kV/cm; 2.5 kV/cm: carrot with PEF 2.5 kV/cm (n = 3).

**Figure 3 foods-12-00589-f003:**
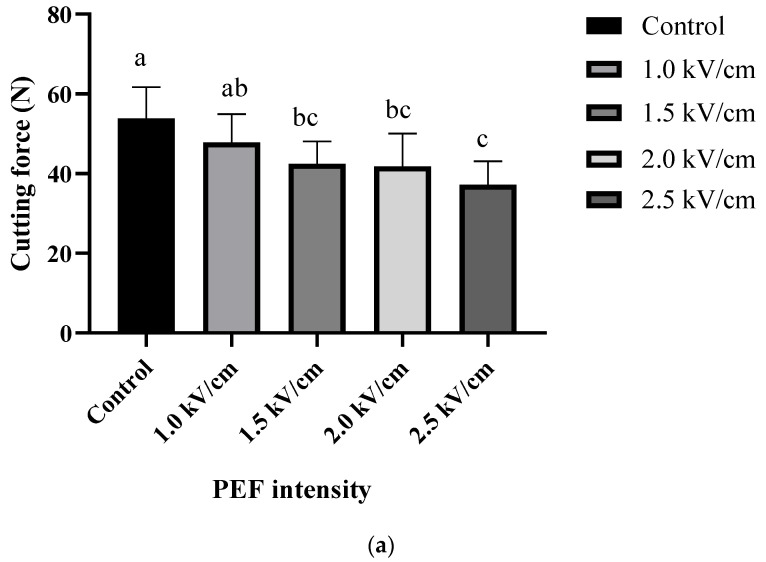
Cutting force (**a**) of carrot after treated by PEF and (**b**) cutting force of carrot after rehydrated at 90 °C. Control: carrot untreated by PEF; 1.0 kV/cm: carrot treated with PEF 1.0 kV/cm, 1.5 kV/cm: carrot treated with PEF 1.5 kV/cm; 2.0 kV/cm: carrot treated with PEF 2.0 kV/cm; 2.5 kV/cm: carrot with PEF 2.5 kV/cm (n = 3). Different letters (a–c) mean significant difference among different samples by Duncan’s multiple range test (*p* < 0.05).

**Figure 4 foods-12-00589-f004:**
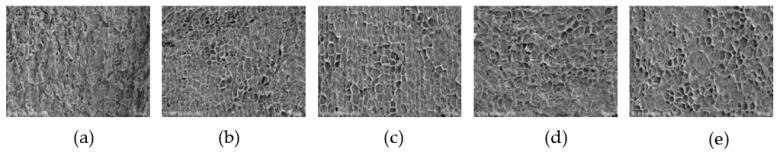
SEM images of carrot with different strengths of PEF pretreatment after rehydration. (**a**) Control: untreated; (**b**) 1.0 kV/cm: carrot treated with PEF 1.0 kV/cm, (**c**) 1.5 kV/cm: carrot treated with PEF 1.5 kV/cm; (**d**) 2.0 kV/cm: carrot treated with PEF 2.0 kV/cm; (**e**) 2.5 kV/cm: carrot with PEF 2.5 kV/cm.

**Figure 5 foods-12-00589-f005:**
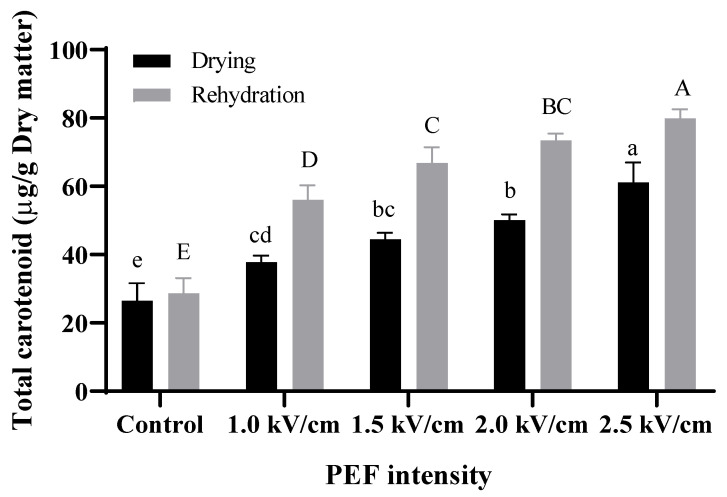
Total carotenoid content of dried carrot with different strengths of PEF pretreatment after dehydrated and rehydrated at 90 °C. Control: untreated; 1.0 kV/cm: carrot treated with PEF 1.0 kV/cm, 1.5 kV/cm: carrot treated with PEF 1.5 kV/cm; 2.0 kV/cm: carrot treated with PEF 2.0 kV/cm; 2.5 kV/cm: carrot with PEF 2.5 kV/cm (n = 3). Values within different lowercase letters (a–d) for drying and uppercase letters (A–C) for rehydration in the same column are significantly different (*p* < 0.05) as determined by Duncan’s test.

**Table 1 foods-12-00589-t001:** Drying model coefficients for selected models.

Drying Model	Newton				Page				Henderson & Pabis			
	Model Parameter	*R* ^2^	RMSE	SSE	Model Parameter	*R* ^2^	RMSE	SSE	Model Parameter	*R* ^2^	RMSE	SSE
Control ^1)^	*k* = 0.5757	0.646	0.131	0.120	*k* = 0.3311 n = 1.0750	0.999	0.005	0.001	a = 1.0869 *k* = 0.3924	0.998	0.009	0.000
1.0 kV/cm	*k* = 0.6009	0.591	0.161	0.134	*k* = 0.2470 n = 1.2929	0.999	0.010	0.001	a = 1.3020 *k* = 0.4537	0.992	0.023	0.004
1.5 kV/cm	*k* = 0.7247	0.603	0.140	0.137	*k* = 0.3746 n = 1.1760	0.979	0.032	0.007	a = 1.5851 *k* = 0.6168	0.858	0.084	0.049
2.0 kV/cm	*k* = 0.7820	0.663	0.137	0.131	*k* = 0.3614 n = 1.2832	0.998	0.010	0.007	a = 1.5192 *k* = 0.6656	0.973	0.039	0.011
2.5 kV/cm	*k* = 0.8495	0.562	0.150	0.157	*k* = 0.3724 n = 1.3000	0.983	0.030	0.007	a = 2.2450 *k* = 0.8117	0.643	0.135	0.128

^1^ Control: untreated; 1.0 kV/cm: carrot treated with PEF 1.0 kV/cm, 1.5 kV/cm: carrot treated with PEF 1.5 kV/cm; 2.0 kV/cm: carrot treated with PEF 2.0 kV/cm; 2.5 kV/cm: carrot with PEF 2.5 kV/cm (n = 3).

**Table 2 foods-12-00589-t002:** Peleg’s model, exponential model, and first-order model constant at different strengths of PEF.

Temp (°C)	Sample	Model Name											
		Peleg’s Model				Exponential Model				First-Order Model			
		Model Parameter	*R* ^2^	RMSE	MRE	Model Parameter	*R* ^2^	RMSE	MRE	Model Parameter	*R* ^2^	RMSE	MRE
90 °C	Control ^1^	*k*_1_ = 0.0815 *k*_2_ = 0.1311	0.9916	0.1109	2.5453	*k*_3_ = 0.9367 *k*_4_ = 0.7074	0.9956	0.0105	2.3746	*k*_3_ =0.8815 *k*_4_ = 1	−209.72	2.3090	585.55
	1.0 kV/cm	*k*_1_ = 0.0800 *k*_2_ = 0.1204	0.9854	0.1615	3.8241	*k*_3_ = 0.8982 *k*_4_ = 0.7211	0.9864	0.0187	4.1473	*k*_3_ = 0.8459 *k*_4_ = 1	−182.17	2.1759	527.12
	1.5 kV/cm	*k*_1_ = 0.0757 *k*_2_ = 0.1198	0.9875	0.1513	3.4958	*k*_3_ = 0.9309 *k*_4_ = 0.7261	0.9856	0.0195	4.4629	*k*_3_ = 0.8731 *k*_4_ = 1	−196.17	2.2773	568.71
	2.0 kV/cm	*k*_1_ = 0.0663 *k*_2_ = 0.1068	0.9927	0.1282	2.7456	*k*_3_ = 0.9382 *k*_4_ = 0.7136	0.9940	0.0124	2.8911	*k*_3_ = 0.8812 *k*_4_ = 1	−206.87	2.3081	583.68
	2.5 kV/cm	*k*_1_ = 0.0572 *k*_2_ = 0.1077	0.9914	0.1385	2.7680	*k*_3_ = 1.0378 *k*_4_ = 0.6984	0.9907	0.0155	3.9789	*k*_3_ = −0.9683 *k*_4_ = 1	−271.32	2.6592	743.28

^1^ Control: untreated; 1.0 kV/cm: carrot treated with PEF 1.0 kV/cm, 1.5 kV/cm: carrot treated with PEF 1.5 kV/cm; 2.0 kV/cm: carrot treated with PEF 2.0 kV/cm; 2.5 kV/cm: carrot with PEF 2.5 kV/cm (n = 3).

## Data Availability

The data are available from the corresponding author.

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
