# Peer review of "A Pulsed Electric Field Accelerates the Mass Transfer during the Convective Drying of Carrots: Drying and Rehydration Kinetics, Texture, and Carotenoid Content"

_foods, 2023, doi:10.3390/foods12030589_

Round 1
Reviewer 1 Report
The research scope is interesting and may find wider interest. However, pulsed electric field treatment is a fairly common research topic undertaken in many articles in recent years. The manuscript concerns the determination of the impact of the pulsating electric field PEF on increasing the efficiency of the drying and re-hydration process of carrots, expressed by changes in the moisture content of the tested material, as well as selected physicochemical properties of the product, i.e. mechanical properties and microstructure, and changes in the content of carotenoids. The most important advantages of the manuscript include the scope of research covering the kinetics of drying and rehydration processes described by means of mathematical equations with the assessment of their fit. Significant flaws are glaring grammatical errors. The manuscript should be substantially revised.
The title includes the topics covered in the manuscript, but is too long and contains a grammatical error; it is worth rewriting it stylistically.
Abstract
Line 12: „We investigate the kinetics of drying and rehydration, texture properties, cellular structure, and carotenoid contents.” The sentence is grammatically incorrect.
“Drying kinetics curves gradually decreased and the lowest drying time required for drying was PEF 2.5 kV/cm at the same time.” The sentence is grammatically incorrect.
The weight difference between Control and PEF 2.5 kV/cm was 27.58% at 60 min during the rehydration process. The sentence is grammatically incorrect; incomprehensible, wrong.
Materials and Methods
Lines 70-71: “.. carrots were cut into 10 mm 69 × 10 mm × 40 mm and stored.” In other parts of the manuscript, carrot samples were prepared in different sizes, which should be stated in the research methodology section.
Lines 84: “..the initial 84 weight was 3.5 g then dried at 50 ℃ using a convective dryer” – Does drying such a small sample give the opportunity to draw appropriate conclusions?
Lines 112-115: It should be justified why the cutting test was used to test the texture of carrot samples. What element and parameters (what knife, what spacing of elements, what percentage of sample damage) were used for this test? Were individual 10x10x40 mm samples tested? How many repetitions were done? Why was the break test not used?
Line 113: “We used a Warner Bratzler reversible blade.” Change the form of the sentence to the passive voice and the past tense here and elsewhere.
137: Duncan’s multiple range test at p<0.05. – should be “at p=0.05”
RESULTS AND DISCUSSION
Lines 148-149: “This polarization effect which associated with the extraction, diffusion, mass transfer, and heat transfer of the food products controlled by pulse number, duration of the pulse, the strength of the field, cell type, and cell structure [23].” – Should extraction be discussed here?
Line 172: “The drying rate with moisture ratio was dramatically decreased in PEF treated group (Figure 2).” - Check and correct this sentence.
Line 215:L "Time for rehydration gradually decreased with an increase in the strengths of PEF." - Here, the processing time was not analyzed, only the rehydration capacity.
Figure 3. On Figs. 1-3 letter designations are of type a, b and here A, B?
Figure 5. In the caption of the figure, under the graph, there should be a type of processing (influence of PEF intensity) rather than "Samples"
With increasing PEF power, the total content of carotenoids in dried carrots increased, which is explained and discussed. However, the correctness of determining the content of these compounds after rehydration should be checked. I am not convinced that the results are correctly specified.
I suggest correcting throughout the manuscript "dehydration" which refers to the removal of water without phase change to "drying" which is more appropriate here.
Conclusions
"Better moisture diffusion of the PEF pretreated group was proved by assessing the drying and rehydration kinetics." - The conclusions mentioned the moisture diffusion coefficient three times, which was not determined in the tested carrot samples..
Author Response
Dear Editor,
The authors are grateful to the editor and reviewers for their interest in the findings and valuable suggestions that helped improve the manuscript. A point-by-point response to the editor’s comments is included below. All changes in the revised manuscript are highlighted in red text font.
|
No. |
question |
line |
→ |
answer |
line |
|
1 |
The title includes the topics covered in the manuscript but is too long and contains a grammatical error; it is worth rewriting it stylistically. |
1 |
We corrected this part “A pulsed electric field accelerates the mass transfer during the convective drying of carrots: drying and rehydration kinetics, texture, and carotenoid content” |
1 |
|
|
2 |
1)We investigate the kinetics of drying and rehydration, texture properties, cellular structure, and carotenoid contents.” The sentence is grammatically incorrect. 2)“Drying kinetics curves gradually decreased and the lowest drying time required for drying was PEF 2.5 kV/cm at the same time.” The sentence is grammatically incorrect. 3)The weight difference between Control and PEF 2.5 kV/cm was 27.58% at 60 min during the rehydration process. The sentence is grammatically incorrect; incomprehensible, and wrong. |
27-33 |
We corrected this part regarding your comments. 1)We investigate the effects of PEF on the rate of drying and rehydration, texture, cellular structure, and carotenoid contents of carrots. 2)The drying rate gradually decreased, and the lowest drying time required for drying was PEF 2.5 kV/cm at the same time. 3)The PEF2.5 kV/cm weight was reduced by 27.58% compared with the Control at 60 min during the rehydration process. |
29-33 |
|
|
3 |
carrots were cut into 10 mm 69 × 10 mm × 40 mm and stored.” In other parts of the manuscript, carrot samples were prepared in different sizes, which should be stated in the research methodology section. |
89-90 |
We corrected this part. we filled in incorrect information about the sample preparation setting. We used samples with 10 mm × 10 mm × 10 mm for all the experiments. |
93 |
|
|
4 |
the initial 84 weight was 3.5 g then dried at 50 ℃ using a convective dryer” – Does drying such a small sample give the opportunity to draw appropriate conclusions? |
108 |
Thank you for your comments. We performed previous experiments using various temperatures (50℃,60℃,70℃,80℃) and then determine the 50℃ to express the drying rate more efficiently. |
|
|
|
5 |
1)It should be justified why the cutting test was used to test the texture of carrot samples.
2)What element and parameters (what knife, what spacing of elements, what percentage of sample damage) were used for this test?
3)Were individual 10x10x40 mm samples tested?
4)How many repetitions were done? Why was the break test not used? |
112-115 |
We corrected this part regarding your comments. 1)The cutting force was measured to evaluate the quality of rehydration. If the rehydration capacity is improved by PEF, less force is required to cutting the carrot due to the intact cellular structure. 2)Cutting force measurements (N) were conducted using a texture analyzer(TAHDi/500; Stable Micro Systems Ltd., Godalming, UK) and based on the maximum shear force (N). A Warner–Bratzler flat blade and a 50-kg power cell were used for the measurements. We corrected this part regarding your comments “We have used a Warner Bratzler reversible blade and a 50-kg power cell was used for the measurements based on the maximum shear force (N).” 3) We corrected this part. we filled in incorrect information about the sample preparation setting. We used samples with 10 mm × 10 mm × 10 mm for all the experiments. 4)For investigating the kinetics of drying and rehydration, 3 points were obtained. And for measuring the texture and carotenoid contents9 repeated measurements were performed, and three experimental results were set to one repetition. We obtained enough data to reach the demand for the statistical analysis. |
142-146 |
|
|
6 |
“We used a Warner Bratzler reversible blade.” Change the form of the sentence to the passive voice and the past tense here and elsewhere. |
132 |
We corrected this part regarding your comments |
146 |
|
|
7 |
Duncan’s multiple range test at p<0.05. – should be “at p=0.05” |
137 |
We corrected this part regarding your comments. |
166 |
|
|
8 |
This polarization effect which associated with the extraction, diffusion, mass transfer, and heat transfer of the food products controlled by pulse number, duration of the pulse, the strength of the field, cell type, and cell structure [23].” – Should extraction be discussed here? |
172 |
We corrected this part regarding your comments. However, we performed quantification of carotenoid contents in this study thus we added this part but rather than write this part in this section, we added this part in the discussion of carotenoid contents. |
178 |
|
|
9 |
“The drying rate with moisture ratio was dramatically decreased in PEF treated group (Figure 2).” - Check and correct this sentence. |
197 |
We corrected this part regarding your comments. The drying rate was decreased in PEF treated group and as the strengths of PEF increased, the drying process was accelerated (Figure 2). |
203 |
|
|
10 |
"Time for rehydration gradually decreased with an increase in the strengths of PEF." - Here, the processing time was not analyzed, only the rehydration capacity. |
242 |
|
We corrected this part regarding your comments. Rehydration capacity was gradually increased with an increase in the strengths of PEF |
249 |
|
11 |
On Figs. 1-3 letter designations are of type a, b and here A, B? |
Fig3 |
|
We corrected this part regarding your comments. |
Fig3 |
|
12 |
In the caption of the figure, under the graph, there should be a type of processing (influence of PEF intensity) rather than "Samples" With increasing PEF power, the total content of carotenoids in dried carrots increased, which is explained and discussed. However, the correctness of determining the content of these compounds after rehydration should be checked. I am not convinced that the results are correctly specified. I suggest correcting throughout the manuscript "dehydration" which refers to the removal of water without phase change to "drying" which is more appropriate here. |
Fig5 |
|
We corrected this part regarding your comments. |
Fig5 |
|
13 |
"Better moisture diffusion of the PEF pretreated group was proved by assessing the drying and rehydration kinetics." - The conclusions mentioned the moisture diffusion coefficient three times, which was not determined in the tested carrot samples. |
384 |
|
We corrected this part regarding your comments. “Better moisture transfer of the PEF pretreated group was proved by assessing the drying and rehydration kinetics.” |
394, 405 |

Reviewer 2 Report
The article entitled Pulsed electric field accelerate the mass transfer during the convective drying of carrots: drying and rehydration kinetics, maintenance of the cellular structure, and carotenoid content do not bring any novelties to the international literature and do not meet the requirements of a highly quoted journal as Food. The number of samples is too low, and the characterization of the process is basic.
Author Response
Dear Editor,
The authors are grateful to the editor and reviewers for their interest in the findings and valuable suggestions that helped improve the manuscript. A point-by-point response to the editor’s comments is included below. All changes in the revised manuscript are highlighted in red text font.
|
Comments |
||||||||||||||||||||||||||||||||||||
|
1) The article entitled Pulsed electric field accelerates the mass transfer during the convective drying of carrots: drying and rehydration kinetics, maintenance of the cellular structure, and carotenoid content do not bring any novelties to the international literature and do not meet the requirements of a highly quoted journal as Food. The number of samples is too low, and the characterization of the process is basic. |
||||||||||||||||||||||||||||||||||||
|
Answer |
||||||||||||||||||||||||||||||||||||
|
Question 1) ->We checked the cited reference again as follows: Table 1: And we corrected revised our manuscript with detailed backgrounds and justifications. Corrected lines are 79-84 as follows: “There has been some previous research on PEF effects on carrot drying using different PEF intensities [11], various pulse numbers and different strengths of PEF (0.5-1.5 kV/cm) [12-15], various pulse duration and pulse repetition time [16]. While few deal with the systematic relationships between PEF effects and drying and rehydration kinetics, texture, microstructure, and carotenoid contents. Furthermore, most of the previous research has been devoted to the PEF effects on the drying properties, therefore there is a paucity of physicochemical, texture, and functional properties regarding after rehydration of carrots.” Table1. The list of research related to PEF and carrot drying
Question 2) Are all the cited references relevant to the research? ->We corrected this part overall again as indicated in question 1. Especially, we added 5 more references in the introduction part [11-16]. Question 3) Are the methods adequately described? -> We corrected several parts regarding your comments. Thank you for your comments. -> [line 96] The electric switch of IGBT (Insulated Gate Bipolar Transistor) and the condition of the electrical connection were the same as followed; power supply voltage: 400V, rated current: 25A, mains frequency: 50Hz and type of connection: 3-phases(3P+PE). -> [line 105] sample, it was cut into 10 mm 10 mm 10 mm then measured range from 1 kHz to 2 MHz then calculated by using Eq.(1). σ(ωs) = 1/A|Z(jω)s| (1) where l, A, and are the length of sample, area horizontal to the electrical field, and system impedance, respectively. Question 4) Are the results clearly presented? -> We corrected several parts regarding your comments. Thank you for your comments. -> [line178] This polarization effect which associated with the diffusion, mass transfer, and heat transfer of the food products controlled by pulse number, duration of the pulse, the strength of the field, cell type, and cell structure [23]. -> [line 203] The drying rate was decreased in PEF treated group and as the strengths of PEF increased, the drying process was accelerated (Figure 2). -> [line 248] Rehydration capacity was gradually increased with an increase in the strengths of PEF. -> [line 283] Disruption of the cell membrane induce the loss of turgor pressure which reduce the hardness of vegetable thus in the current study, the decrease in cutting force of carrot is influenced by the rupture of the cell membrane. Question 5) Are the conclusions supported by the results? -> [line 404] These results indicate that PEF treatment can serve as a pretreatment method to accelerate the moisture transfer during drying and rehydration and to modify the physicochemical properties of dried carrot, which include tissue cutting force, cellular structure, and carotenoids contents, by inducing electroporation on cell membrane. Comments 6) The number of samples is too low, and the characterization of the process is basic. ->For investigating the kinetics of drying and rehydration, 3 points were obtained. And for measuring the texture and carotenoid contents, 9 repeated measurements were performed, and three experimental results were set to one repetition. We obtained enough data to reach the demand for the statistical analysis. Thank you for your comments. We sincerely revised our manuscript. |

Round 2
Reviewer 1 Report
As I wrote in a previous review, the manuscript is valuable. Although the use of PEF treatment is quite extensively researched on a laboratory scale, and the results of the research are available in many articles, these studies confirm the usefulness of this treatment and may be transferred to a wider scale. A particularly important application of PEF can be used to shape the structure and behavior of biocomponents such as carotenoids. Although the manuscript has been significantly improved, it is still not good enough. The abstract was particularly carelessly prepared. Also, the applications do not contain specific information, e.g. by how much % the drying time was reduced and how well the carotenoids were preserved, compared to the raw material and the control sample.
The abstract as a showcase of the manuscript is not well corrected. The abstract is still not well written, it does not relate to the results discussed in the manuscript, and it does not relate to the stated aim. The sentences are incomprehensible. There is no specific information on the effect of PEF on the drying and rehydration process and on enhancing the behavior of carotenoids during convection drying. The final conclusion is like an abstraction.
“We investigate the effects of PEF on the rate of drying and rehydration, texture, cellular structure, and carotenoid contents of carrots.” - Grammatically incorrect sentence.
“The drying rate gradually decreased, and the lowest drying time required for drying was PEF 2.5 kV/cm at the same time.” - "The drying rate gradually decreased" - Under what conditions? Is it about increasing the strength of PEF or extending the time?
"the lowest drying time required for drying was PEF 2.5 kV/cm at the same time" - What does it mean?
"The PEF2.5 kV/cm weight was reduced by 27.58% compared with the Control at 60 min during the rehydration process." - This is a badly worded sentence. During rehydration, the weight of the samples increases. What does "reduced by 27.58%" mean? Did PEF make the samples less hydrated?
“Due to the reduction in exposure time to oxygen and hot air, the preservation of carotenoid content was increased depending on the strength of PEF.” - How much has the drying time (meaning "the reduction in exposure time to oxygen and hot air"?) been shortened? How much carotenoids have been retained?
Introduction
"Convective drying is the most economical thermal technology which is uses passing air and a regulated relatively high temperature" - This is not true. If so, why use treatments that improve drying, such as PEF?
Result
Check the range of PDF processing frequency used in the methodology and in Fig. 1a.
Check if "Frequency log KHz" is correct or should it be "Frequency log kHz"?
Why only in Fig. 5 is the axis caption: "PEF intensity"
The drying rate was decreased in PEF treated group and as the strengths of PEF increased, the drying process was accelerated (Figure 2)." - How did the PEF treatment affect the drying rate? However, in graph 2a there is no "drying rate", there is simply drying time.
“The drying rate was decreased in PEF treated group and as the strengths of PEF increased, the drying process was accelerated (Figure 2).” - To determine the impact of PEF, authors should specify specific values expressed as a difference in drying time (no drying rate!) or as a percentage, comparing the drying time of individual samples!
Author Response
|
Journal |
Foods |
|
Title |
A pulsed electric field accelerates the mass transfer during the convective drying of carrots: drying and rehydration kinetics, texture, and carotenoid content |
Dear Editor,
The authors are grateful to the editor and reviewers for their interest in the findings and valuable suggestions that helped improve the manuscript. A point-by-point response to the editor’s comments is included below. All changes in the revised manuscript are highlighted in red text font.
|
No. |
question |
line |
→ |
answer |
line |
|
1 |
The abstract as a showcase of the manuscript is not well corrected. The abstract is still not well written, it does not relate to the results discussed in the manuscript, and it does not relate to the stated aim. The sentences are incomprehensible. There is no specific information on the effect of PEF on the drying and rehydration process and on enhancing the behavior of carotenoids during convection drying. The final conclusion is like an abstraction. |
30 |
We revised this part regarding your comments. Thank you. è “In this study, the drying and rehydration kinetics, microstructure, and carotenoid content of carrot (Daucus carota) pretreated by PEF during the convective drying at 50℃ were investigated. The PEF treatment was conducted with different field strengths (1.0-2.5 kV/cm) using a fixed pulse width of 20 µs and at a pulse frequency of 50 Hz. The PEF 2.5 kV/cm showed the shortest drying time, taking 180 min, whereas the control required 330 min for the same moisture ratio, indicating a 45% reduction in drying time. The rehydration ability also increased as the strengths of PEF increased. PEF 2.5 kV/cm resulted in 27.58% increase in moisture content compared to Control after rehydration (1h). Three mathematical models were fitted to the drying and rehydration data, Page and Peleg model were selected as the highest fit models to describe the drying and rehydration kinetics, respectively. The cutting force of the sample was decreased as the strength of PEF increased and a more homogeneous cellular structure was observed in the PEF pretreatment group. The reduction in drying time by PEF was beneficial to the carotenoid content, and PEF2.5kV/cm showed the highest preservation content of carotenoid. Overall, these results suggested that the pretreatment of PEF and the drying and rehydration rate influence the quality of products, functional components, and cellular structure.” |
27-41
|
|
|
2 |
“We investigate the effects of PEF on the rate of drying and rehydration, texture, cellular structure, and carotenoid contents of carrots.” - Grammatically incorrect sentence. |
27-33 |
We revised this part regarding your comments. Thank you. è “In this study, the drying and rehydration kinetics, microstructure, and carotenoid content of carrot (Daucus carota) pretreated by PEF during the convective drying at 50℃ were investigated.” |
27 |
|
|
3 |
“The drying rate gradually decreased, and the lowest drying time required for drying was PEF 2.5 kV/cm at the same time.” - "The drying rate gradually decreased" - Under what conditions? Is it about increasing the strength of PEF or extending the time? "the lowest drying time required for drying was PEF 2.5 kV/cm at the same time" - What does it mean?
|
31 |
We added more information about drying condition regarding your comments. Thank you. è “The PEF 2.5 kV/cm showed the shortest drying time, taking 180 min, whereas the control required 330 min for the same moisture ratio, indicating a 45% reduction in drying time.” è What we were trying to say was, "The PEF-2.5kV/cm had the lowest moisture content at the same time period compared to Control." |
31 |
|
|
4 |
"The PEF2.5 kV/cm weight was reduced by 27.58% compared with the Control at 60 min during the rehydration process." - This is a badly worded sentence. During rehydration, the weight of the samples increases. What does "reduced by 27.58%" mean? Did PEF make the samples less hydrated? conclusions? |
33 |
“Among them, the time with the highest difference in weight was 60 min after rehydration, and there was a 27.58 % higher in PEF 2.5 kV/cm compared to the control.” To be specific, we revised this part regarding your comments. è “The rehydration ability also increased as the strengths of PEF increased. PEF 2.5 kV/cm resulted in 27.58% increase in moisture content compared to Control after rehydration (1h).” |
33 |
|
|
5 |
“Due to the reduction in exposure time to oxygen and hot air, the preservation of carotenoid content was increased depending on the strength of PEF.” - How much has the drying time (meaning "the reduction in exposure time to oxygen and hot air"?) been shortened? How much carotenoids have been retained? |
35 |
è The drying time of the sample ranged from 180 min to 330 min. Equilibrium moisture content was reached after 5.5 h (Control), 4 h (PEF1.0), 3.5 h (PEF1.5), and 3 h (PEF2.0 and PEF 2.5), respectively approximately more than 45% drying time was reduced compared to Control. è And “The carotenoid contents of Control, PEF 1.0 kV/cm, PEF 1.5 kV/cm, PEF 2.0 kV/cm, and PEF 2.5 kV/cm were 26.15±5.12 μg/g, 37.82±1.87 μg/g, 44.44±1.90 μg/g, 50.03±1.81 μg/g and 61.12±5.84 μg/g, respectively in dried carrot.” To be specific, we revised this part regarding your comments. è “The reduction in drying time by PEF was beneficial to the carotenoid content, and PEF2.5kV/cm showed the highest preservation content of carotenoid.” |
38 |
|
|
6 |
"Convective drying is the most economical thermal technology which is uses passing air and a regulated relatively high temperature" - This is not true. If so, why use treatments that improve drying, such as PEF? |
60 |
Convective drying is a representative thermal technology of drying using relatively strong heat in the food industry especially for drying croppers. Due to the strong heat, surface drying occurs quickly, and the internal moisture cannot be removed homogeneously, causing surface hardening and nutrient loss, sometimes requiring prolonged drying. To improve this problem, pulsed electric fields were used to increase cell membrane permeability, allowing moisture to be removed homogeneously and uniformly thus reducing the side effects caused by convective drying. |
61 |
|
|
7 |
Check the range of PDF processing frequency used in the methodology and in Fig. 1a. |
194 |
We checked and revised this part regarding your comments. |
195 |
|
|
8 |
Check if "Frequency log KHz" is correct or should it be "Frequency log kHz"? |
194 |
We checked and revised this part regarding your comments. |
195 |
|
|
9 |
Why only in Fig. 5 is the axis caption: "PEF intensity" |
Figure |
We checked and revised all Figures regarding your comments. |
Figure |
|
|
10 |
The drying rate was decreased in PEF treated group and as the strengths of PEF increased, the drying process was accelerated (Figure 2)." – 1) How did the PEF treatment affect the drying rate? 2) However, in graph 2a there is no "drying rate", there is simply drying time. |
203 |
|
è The pretreatment of PEF increases the permeability of the cell membrane. The permeability of the cytoplasm membrane, which plays a major role as a regulator of mass transfer, increased thus enhancing the diffusion and migration of small molecules. Therefore, the drying time accelerated. We checked and revised this part regarding your comments. è “The drying time was decreased in PEF treated group and as the strengths of PEF increased, the drying process was accelerated (Figure 2).” |
204 |
|
11 |
“The drying rate was decreased in PEF treated group and as the strengths of PEF increased, the drying process was accelerated (Figure 2).” - To determine the impact of PEF, authors should specify specific values expressed as a difference in drying time (no drying rate!) or as a percentage, comparing the drying time of individual samples! |
Fig3 |
|
We fixed based on drying time and revised all parts regarding your comments. ð “The drying time was decreased in PEF treated group and as the strengths of PEF increased, the drying process was accelerated (Figure 2).” ð The drying time of the sample ranged from 180 min to 330 min. Equilibrium moisture content was reached after 5.5 h (Control), 4 h (PEF1.0), 3.5 h (PEF1.5), and 3 h (PEF2.0 and PEF 2.5), respectively approximately more than 45% drying time was reduced compared to Control. |
204. 206 |

Reviewer 2 Report
Accept as it is.
Author Response
Thank you for your comments and evaluation.
Thank you.